

# Generating, retrieving persona and generating responses for long-term open-domain dialogue

Dohyun Cha[1], Dawon Lee[2] and Jihie Kim[1]

[1] Department of Computer Science and Artificial Intelligence, Dongguk University, Seoul, Republic of South Korea
[2] Division of AI Software Convergence, Dongguk University, Seoul, Republic of South Korea

## ABSTRACT

Open-domain dialogue systems have shown remarkable capabilities in generating natural and consistent responses in short-term conversations. However, in long-term conversations such as multi-session chat (MSC), where the dialogue history exceeds the model's maximum input length (*i.e.*, 1024 tokens), existing dialogue generation systems often overlook the information from earlier dialogues, leading to the loss of context. To prevent such loss and generate natural, consistent responses, we propose a GRGPerDialogue framework, consisting of three main stages: generating persona from past dialogues, retrieving persona relevant to the current utterance, and generating responses based on both persona and recent dialogues. In the first stage, we generate the persona of each speaker in real-time with diverse expressions, leveraging Llama 2 In-Context Learning (ICL). Subsequently, we propose a new dataset called Persona-Utterance Pair (PUP) and use it to train Facebook dense passage retrieval (DPR) model for retrieving persona sentences relevant to the current utterance. Finally, we train generative models such as Generative Pre-trained Transformer 2 (GPT-2) and Bidirectional and Auto-Regressive Transformers (BART) to generate responses based on retrieved persona sentences and the recent dialogues. Experimental results on a long-term dialogue dataset demonstrate that the GRGPerDialogue framework outperforms baseline models by approximately 0.6% to 1% in terms of the Rouge-1 metric. Furthermore, human evaluation results supported the effectiveness of GRGPerDialogue. These results indicate that GRGPerDialogue can generate responses that are not only more fluent and consistent, but also more relevant to the dialogue history than baseline models.

## INTRODUCTION

The objective of open-domain dialogue is to enable conversational systems to engage in natural conversations with humans across a wide range of topics and situations (*Clark et al., 2019*; *Roller et al., 2020*; *Zhou et al., 2021*; *Sun & Ding, 2022*). Recently, transformer-based generative pre-trained language models have enabled dialogue systems to generate responses that are natural and consistent. However, this approach primarily applies to

Corresponding author
Jihie Kim, jihie.kim@dgu.edu

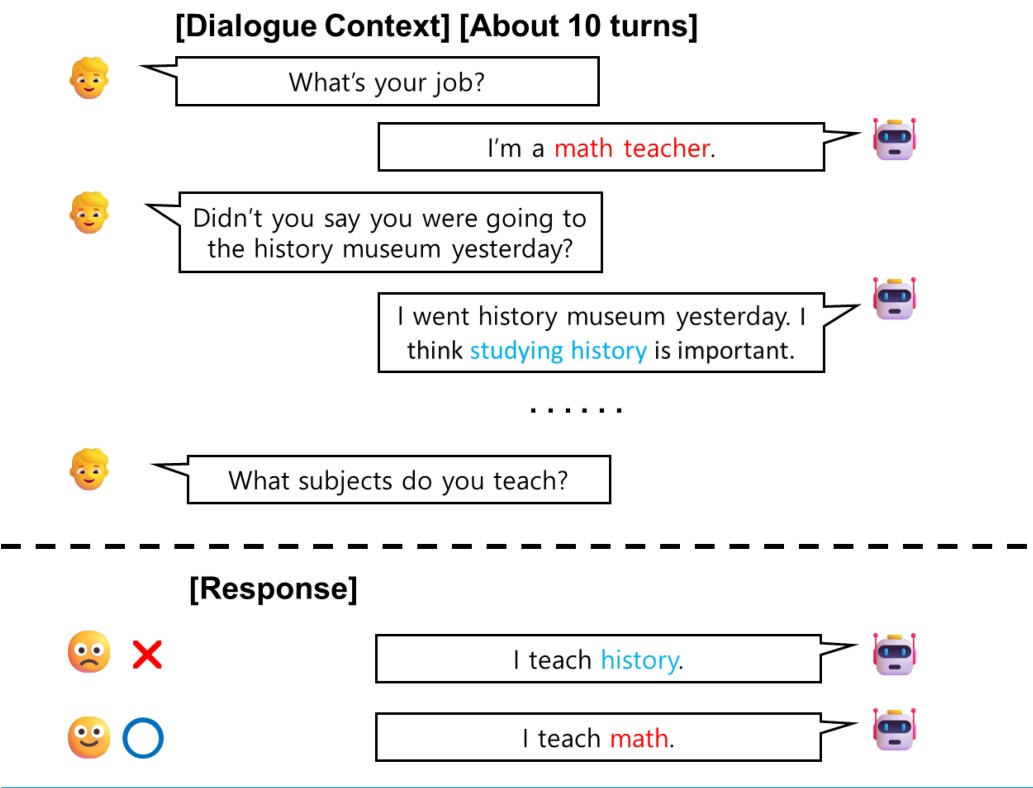

**Figure 1  An illustrated example of a multi-turn dialogue between a user and an agent.** Red text indicates information necessary for generating a response, while blue text represents irrelevant information that could confuse the model. The Agent (GPT2) inferred incorrect information and provided inconsistent responses.

models that support large input sequence lengths. As conversation becomes longer, there will be a point where using the entire history becomes impractical. Therefore, some models have used a truncation to make use of the most recent conversations as input when generating responses, which results in a loss of information in earlier conversations. For example, the multi-session chat (*Xu, Szlam & Weston, 2022*) is a long-term dialogue dataset consisting of dialogues with five sessions, each containing approximately 12 utterances. Therefore, there are cases where the model cannot utilize the information from previous sessions in the dialogue. Another issue when using the entire history is that the model can encounter difficulties in finding necessary information from long sequences. As illustrated in Fig. 1, since the information in dialogues is scattered, models frequently infer incorrect historical information, leading to inconsistent responses.

Previously, various methods have been developed to achieve better results in long-term dialogue. One line of them is the dialogue summarization method (*Xu, Szlam & Weston, 2022*; *Zhong et al., 2022*), where a summarization module condenses the dialogue history into a shorter form for use in response generation. However, these methods often rely on manually summarized data and lack clear criteria for selecting which information should be included, which can lead to inconsistent or suboptimal summaries. The other line is also called retrieval-augmented method (*Ramakrishnan et al., 2022*; *Han et al., 2021*; *Shu*

*et al., 2023*). It involves retrieving parts of the context relevant to the current utterance. However, since this method uses parts of the dialogue or utterance verbatim, it fails to find information that is not explicitly stated in the dialogue.

In this article, we aim to address the following research question: "*Which information within the dialogue history is helpful for generating responses in open-domain dialogue?*" We hypothesize that a comprehensive understanding of a speaker's 'persona'—which reflects both static and dynamic aspects of their identity and conversational context—can aid in generating more coherent and contextually appropriate responses in open-domain dialogue. To clarify, we define 'persona' in this work as a structured representation of a speaker's profile, which includes both static elements such as age, occupation, or other enduring characteristics, and dynamic elements such as preferences (*e.g.*, what they like) and behaviors (*e.g.*, what they do in specific situations). Therefore, we summarize previous dialogues into the persona of each speaker while retaining some parts of the recent dialogue for dialogue generation. Unlike a fixed personal profile, the persona is dynamically generated from the dialogue history. Each specific aspect of the persona is captured as an individual persona sentence, providing granular insights into the speaker's identity and interaction style.

This study proposes a GRGPerDialogue framework, consisting of three steps: (1) generating persona from previous sessions, (2) retrieving generated persona sentences relevant to the current utterance, and (3) generating responses using retrieved persona sentences and the current session. We utilize the widely-used Multi-Session Chat dataset, which consists of long-term dialogues with multiple sessions and provides pre-defined personas. Instead of using pre-defined personas, we generate each speaker's persona from previous sessions using Llama 2 (*Touvron et al., 2023*). By leveraging Llama 2's powerful language modeling capabilities, we can generate diverse and nuanced expressions of persona sentences. This approach enables generalization for other datasets that do not provide persona information. Then, we propose a persona retrieval model to retrieve generated persona sentences relevant to the current utterance. To fine-tune the persona retrieval model, we propose a new dataset called Persona-Utterance Pair (PUP). This dataset consists of paired data of utterances and persona sentences relevant to utterances, allowing each to be trained on their respective encoders. Finally, we propose a model to generate responses by concatenating retrieved persona sentences with utterances from the current session.

To validate the effectiveness of our GRGPerDialogue framework in persona generation and persona retrieval, we apply the GRGPerDialogue framework to generative models including Generative Pre-trained Transformer 2 (GPT-2) (*Radford et al., 2019*) and Bidirectional and Auto-Regressive Transformers (BART) (*Lewis et al., 2020*). That is, our baseline models involve fine-tuning GPT2 and BART by concatenating the dialogue as much as possible without exceeding the maximum input sequence length for the model. Experimental results on Facebook MSC dataset show that the GRGPerDialogue framework outperforms baseline models, such as fine-tuned GPT2 and BART, across diverse evaluation metrics. Human evaluation results show that GRGPerDialogue generates more fluent, history-relevant, and consistent responses than baseline models. In addition,

when evaluated with the Conversation Chronicles dataset (*Jang, Boo & Kim, 2023*), which does not provide pre-defined personas, the proposed model achieved better performance than baseline models.

The main contributions of this work are:

(1) We propose a new GRGPerDialogue framework that can generate each speaker's persona from past dialogues and retrieve persona sentences relevant to the current utterance to aid in response generation. In particular, we find that the persona generated in real-time during the persona generation stage of the GRGPerDialogue framework can summarize dialogue history.

(2) Within the GRGPerDialogue framework, we also propose a retrieval model to retrieve persona sentences relevant to the current utterance and propose a new PUP dataset for model fine-tuning.

(3) Automatic and human evaluations show that our framework outperforms baseline models, indicating that summarizing old dialogues into persona while retaining some parts of recent dialogues can contribute to better response generation. Furthermore, we found that using the GRGPerDialogue framework achieves better results than baseline models, even in long-term open-domain dialogue datasets where personas are not provided.

## RELATED WORK

### Response generation in long-term dialogue

The purpose of open-domain dialogue systems is to facilitate natural and engaging conversations between humans and machines across a wide range of topics and contexts. Recently, the advancement of open-domain dialogue systems has been propelled by availability of large-scale dialogue datasets and the development of transformer-based language models, such as Meena (*Adiwardana et al., 2020*), DialogueGPT (*Zhang et al., 2020*), BlenderBOT (*Roller et al., 2021*), and PLATO (*Bao et al., 2020*), trained on large-scale corpora. Despite these advancements, recent models are constrained by the maximum input length, which is typically 1,024 tokens, preventing them from fully utilizing historical contexts in multi-session conversations such as MSC. To address this limitation, recent approaches have employed techniques such as summarizing previous dialogues or retrieving past contexts.

In the case of summarization models, manually summarized data are required, making it challenging to choose which elements should be included in the summary. Long dialogue summarization datasets are more difficult to collect as they consist of thousands of input words, which still necessitate long input sequences (*Zhong et al., 2021*; *Chen et al., 2022*). Recently, a method for automatic summarization with large language models (LLMs) has also been proposed (*Wang et al., 2023*). However, human evaluation is still required for verification of hallucinations and consistency (*Vakharia et al., 2023*).

Another case of retrieval-based models involves retrieving sentences relevant to the current utterance within the dialogue history to generate responses (*Ramakrishnan et al., 2022*; *Han et al., 2021*; *Shu et al., 2023*). These models rely on extracting specific pieces of information from the dialogue history that are directly related to the current context,

aiming to improve the relevance and coherence of the generated responses. However, this approach is inherently limited by its dependence on explicit information present in the dialogue history. If the required context or details are implicit or scattered across multiple turns of the conversation, the retrieval model may struggle to accurately identify and aggregate the necessary information. Additionally, navigating extensive dialogue histories increases computational complexity and poses challenges in efficiently retrieving the most pertinent information. For instance, *Chen et al. (2024)* highlight that retrieval-based methods face difficulties in managing growing memory databases and ensuring the retrieval of relevant memories, especially as conversations accumulate over time.

## Use of persona in dialogue generation

In open-domain dialogue research, generative models are trained on a vast amount of diverse utterances from many individuals. Consequently, the model encounters the issue of not generating consistent sentences (*Li et al., 2016*). To address this issue, dialogue datasets containing pre-defined personas, such as PersonaChat (*Zhang et al., 2018*) and Multi-Session Chat (*Xu, Szlam & Weston, 2022*), were proposed. However, using pre-defined personas cannot be dynamically updated during the chat. Moreover, creating such datasets requires significant resources. Therefore, recent research has focused on methods enabling models to dynamically extract or infer personas from dialogue history. *Wu et al. (2020)* introduced a two-stage attribute extractor, which utilizes the Persona Chat dataset and the Dialogue Natural Language Inference (DialogueNLI) dataset (*Welleck et al., 2019*) to extract persona attribute triplets from dialogues. However, *Zhu et al. (2023)* identified inconsistencies and ambiguities in the annotations of the DialogueNLI dataset. To address this, they proposed PersonaEXT, a dataset with better-annotated data. Nevertheless, PersonaEXT lacked a robust framework for handling implicit persona attributes and did not fully leverage contextual information from ongoing utterances during persona extraction. PLATO-LTM (*Xu et al., 2022*) extracted persona sentences from each utterance and utilized them to generate responses. However, this method also limits the extraction of persona that is not explicitly manifested in utterances. To obtain the persona not explicitly manifested, *Wang et al. (2022)* used GPT2 to infer the persona. This method also restricts the ability to infer personas with diverse expressions due to its reliance on a fixed set of entities and relation types. To overcome these limitations mentioned above, we use the ability of Llama 2 in-context learning to generate the persona of each speaker in real-time with diverse expressions.

## Persona retrieval in dialogue generation

In persona dialogue research, two methods are commonly used for incorporating persona into response generation. The first method involves inputting all persona sentences into the generation model and handling them internally within the model. One of them applies an adaptive attention mechanism within the generation model to integrate information between persona and context (*Huang et al., 2023*). The second method employs a retrieval model to fetch relevant persona sentences. Modular Prompted Chatbot (*Lee et al., 2023*) employs the DPR model to find persona-like information relevant to the current utterance.

However, the DPR model used in the previous study (*Lee et al., 2023*) was trained on pairs of Wikipedia passages and utterances (*Feng et al., 2021*), making it unsuitable for retrieving relevant persona sentences. One study (*Xu et al., 2022*) proposed the DuLeMon dataset, which annotates the persona relevant to the response. They trained their retrieval model using the DuLeMon dataset to retrieve relevant persona sentences. However, the DuLeMon dataset is in Chinese. Annotating it also takes a considerable amount of time. In this article, we propose a new dataset called PUP, created with minimal use of human annotation. Additionally, we train the model using this dataset to retrieve persona sentences necessary for response generation.

## METHODOLOGY

In this section, we introduce the GRGPerDialogue framework aimed at validating our hypothesis: "Summarizing previous dialogues into each speaker's persona while retaining recent dialogues will aid in response generation". As shown in Fig. 2, the GRGPerDialogue framework consists of three steps: persona generation, persona retrieval, and response generation. First, we describe the dataset used in our work. Next, we explain the method of persona generation using Llama 2 in-context learning. Then, we detail the construction of the PUP dataset and the training of the persona retrieval model based on it. Finally, we describe the process of generating responses.

### Multi-session chat

A conversation in the multi-session chat (MSC) (*Xu, Szlam & Weston, 2022*) consists of the current session, representing the ongoing conversation, and several previous sessions. We denote the MSC dataset $D$ as a list of $N$ conversations in the format of ($H$, $C$, $R$). Here, $H$ = $H^1, H^2, \cdots, H^M$ and $H^M$ denotes the conversation of the M-th session. $H^i = h_1^i, h_2^i, \cdots, h_l^i$, where each element represents individual history utterances. $C = c_1, c_2, \cdots c_n$. It represents $n$ utterances of the current conversation session, where $c_n$ denotes the current utterance. Our task is to generate persona from each element in all $H^i$, retrieve relevant persona sentences from the generated persona list, concatenate them with the current session, and generate response $R$.

### Persona generation

To summarize long-term dialogues from previous sessions into persona sentences, we generate the persona of each speaker in the dialogue using Llama 2 in-context learning capability. Although the GPT-3.5 model may generate persona effectively, we decided to use the freely available Llama 2 model considering the subscription cost for the GPT-3.5 model. First, we define the persona to be generated from the dialogue using the format "subject is related with object". In this structure, "is related with" serves as a placeholder for verbs that describe the nature of the interaction or connection, such as "has", "likes", "doesn't have", or "went". This flexibility allows the format to encompass various types of relationships or actions expressed in the dialogue. The purpose of using this standardized format is to simplify the data refinement and management process during post-processing by reducing persona sentences into a unified structure. This format is also included in the prompt

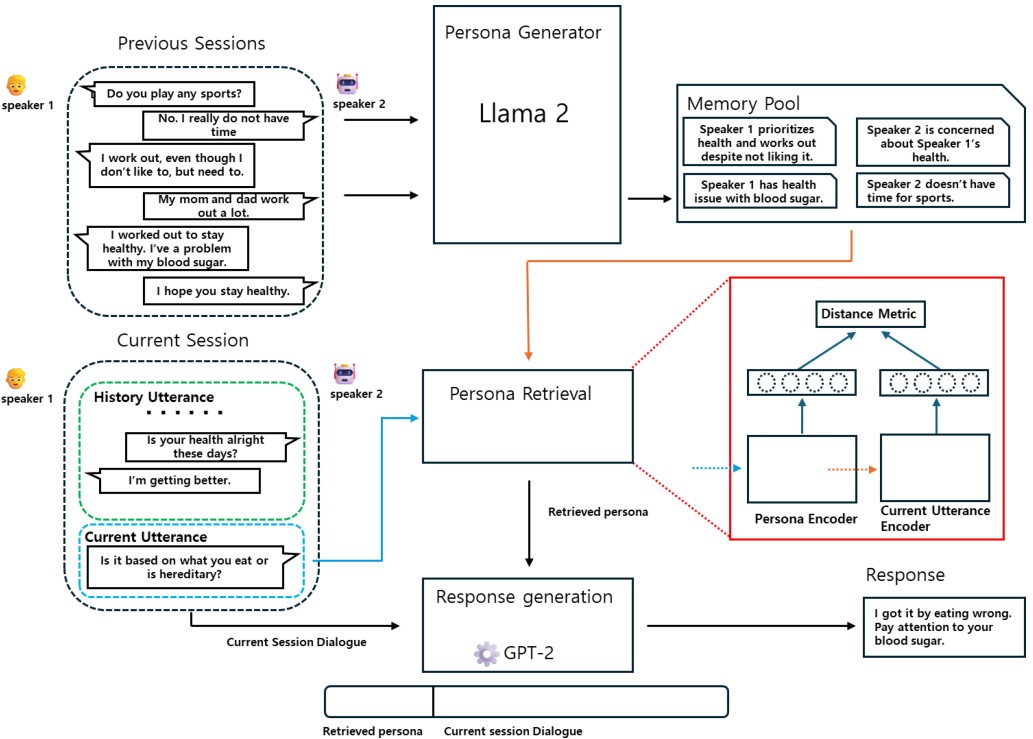

**Figure 2** **Overview of our framework GRGPerDialogue, consisting of a persona generator, persona retrieval, and a response generator.** (A) The persona generator generates the persona from previous sessions and saves them in a memory pool. (B) The persona retrieval retrieves persona sentences relevant to the current utterance from the persona list stored in the memory pool. (C) The response generator generates a response concatenating the retrieved persona sentences and the current session's dialogue.

to guide the persona generation process. Next, we direct the Llama 2 model to generate persona for the previous session, using the persona directly generated by humans from the sample dialogue as a one-shot example. The reason for including only one example is that more examples could lead to exceeding the maximum sequence length of Llama 2, which is 1,024 in our settings. Finally, after generating the persona from every previous session dialogue with Llama 2, we proceed with post-processing. Since Llama 2 is a generative model, unnecessary sentences or words such as "thanks" might be generated along with the persona. These unnecessary sentences or words need to be removed for the next step, which is the persona retrieval model; we utilize regular expressions to remove them from the generated sentences. All processes described above, including actual prompts used, can be seen in Fig. 3.

## Persona retrieval

Instead of using all generated persona sentences from the list, we retrieve a few persona sentences that are most relevant to the current utterance. Initially, we approached this task as a multiple-choice problem where single or multiple correct responses (*Liu et al., 2018*) were selected from a list of the generated persona. We provided utterances and generated persona sentences to Llama 2, prompting it to select relevant options. However, in all

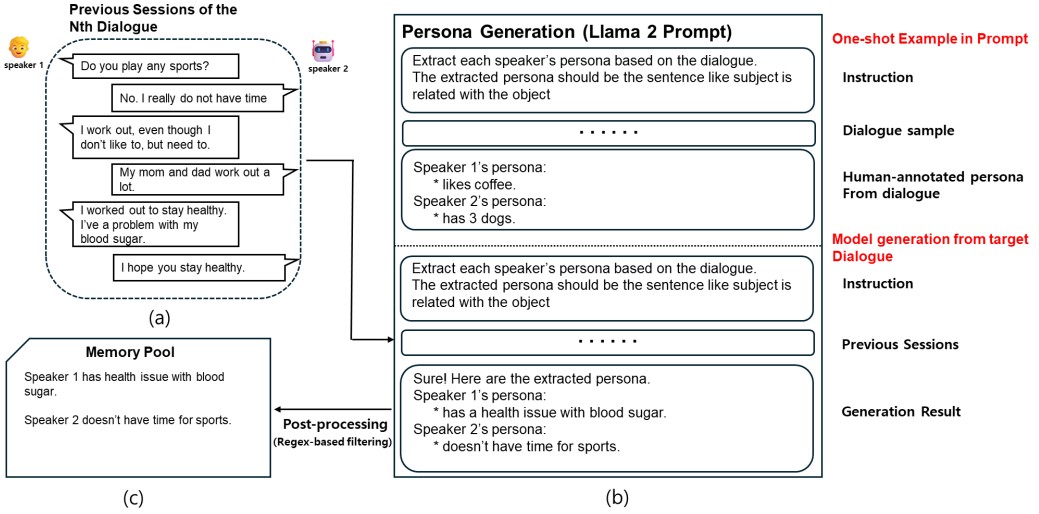

**Figure 3  A workflow of persona generation using in-context learning.** (A) The dialogue history used as input. (B) The prompt structure given to Llama 2, which includes a one-shot example (top) and the actual input dialogue (bottom). The model generates persona sentences for each speaker. A post-processing step using regex-based filtering is applied to remove irrelevant or generic statements. The refined personas are stored in the memory pool shown in (C).

k-shot settings ($0 \leq k \leq 3$), we notice that the chosen response by Llama 2 changes when choices are rearranged, and it frequently provides incorrect answers each time.

**PUP (Persona-Utterance Pair Dataset).** We construct a new PUP dataset, which consists of pairs of utterances and personas related to the utterance. We then fine-tune the Bi-Encoder model on the PUP dataset to retrieve relevant persona sentences. To construct the PUP, we first group consecutive utterances in the conversation into pairs as $(c_i, c_{i+1})$. Next, using Llama 2, we generate persona $p_{i+1}$ from utterance $c_{i+1}$ to construct pairs $(c_i, p_{i+1})$. Finally, we apply regular expressions during post-processing to remove sentences or words that are irrelevant to the generated persona. Subjects "speaker 1" and "speaker 2" are also removed to address the issue of high similarity caused by these subjects during subsequent training or inference processes. The aforementioned processes are applied to the MSC session 1, resulting in the generation of over 47,000 positive pairs. Of these positive pairs, 80% are used as the training set and, 20% are used as the validation set.

**Training procedure.** The dense passage retrieval (DPR) model (*Karpukhin et al., 2020*) proposed by Facebook adopts a bi-encoder architecture. It compares similarity between a question and a Wikipedia passage to search for passages relevant to the question. Inspired by DPR, we use Current Utterance Encoder $E_c(\cdot)$ to encode the utterance $c_i$, and use Persona Encoder $E_p(\cdot)$ to encode the persona $p_{i+1}$.

Encoder $E_c$ and $E_p$ are initialized with the standard BERT pre-trained model (*Devlin et al., 2019*) and then fine-tuned on our PUP dataset. For each training sample, we define the positive pair $p_{i+1}^+$ as generated persona from the next utterance and negative pair $p_{i+1}^-$ as sampled randomly from persona generated of different dialogues. Given the utterance $c_i$, positive persona $p_{i+1}^+$, and negative persona $p_{i+1}^-$ are used to tune our network. We optimize

the loss function as the negative log-likelihood of the positive persona:

$$L(c_i, p_{i+1}^+, p_{i+1,1}^-, \ldots, p_{i+1,n}^-) = -\log\left(\frac{e^{\text{sim}(c_i, p_{i+1}^+)}}{e^{\text{sim}(c_i, p_{i+1}^+)} + \sum_{j=1}^n e^{\text{sim}(c_i, p_{i+1,j}^-)}}\right).$$

We define similarity as the dot product of each encoder's output:

$$\text{sim}(c, p) = E_c(c)^T E_p(p).$$

**Inference procedure**. Once the persona generation process is complete, a persona list $L_p = \rho_1, \rho_2, \cdots, \rho_n$ containing n persona sentences is generated from the history session. Since the persona list contains persona sentences for both Speaker 1 and Speaker 2, we separate them into two lists: $L_{sp1}$ and, $L_{sp2}$. During the inference time, when the speaker of the current utterance is Speaker 1, we select the persona list $L_{sp2}$, and vice versa. Therefore, when the speaker of the current utterance is Speaker 1, the Current Utterance Encoder encodes the current utterance, while the Persona Encoder dynamically encodes each element of the persona list $L_{sp2}$. In the end, we utilize a threshold value $p_{thres}$ to select all persona sentences for which the dot product result between $E_c(c)$ and $E_p(p)$ exceeds the threshold. These selected persona sentences are then added to the list $L_{rt}$. After conducting a few experiments to compare different threshold values, we found that setting $p_{thres} = 0.68$ achieved the best performance. This value showed the highest similarity to the ground truth based on ROUGE-N and BLEU-N scores. Lower threshold values tended to include irrelevant personas, while higher values missed potentially useful personas.

## Response generation

During training, when a relevant persona list $L_{rt}$ is retrieved along with the current utterance, the model generates a response based on persona sentences and current session dialogue. To separate each persona sentence, we use '|' as a separator and insert the special token $<sep>$ between persona sentences and the current utterance. Within the utterance, we use special tokens $<sp1>$ and $<sp2>$ to distinguish between speakers. With the input format mentioned above, encoder encodes them into a hidden vector $h_n$:

$$h_n = Enc(Emb(L_{rt}; c_1; c_2; \cdots; c_n)).$$

The response decoder, taking $h_n$ as input, generates the response $\hat{R}_t$ token by token:

$$P(\hat{r}_{t,k}) = Dec(\hat{r}_{t,k} | \hat{R}_{t<k}, h_t).$$

The cross entropy loss between the generated response $\hat{R}_t$ and the ground truth response $R_t$ is calculated as:

$$\mathcal{L} = \sum_{R_t \in D} CELoss(\hat{R}_t, R_t).$$

## EXPERIMENTAL SETTINGS

In this section, we will introduce the experimental dataset, implementation details, baseline methods, and evaluation metrics.

**Table 1  Statistics of multi-session chat and conversation chronicles datasets.**

| (a) Multi-Session Chat | | | | | |
|---|---|---|---|---|---|
| | Train | | Valid | | Test |
| Type | Ep. | Ut. | Ep. | Ut. | Ut. |
| S1 | 8,939 | 131,438 | 1,000 | 7,801 | 6,634 |
| S2 | 4,000 | 46,420 | 500 | 5,897 | 5,939 |
| S3 | 4,000 | 47,259 | 500 | 5,890 | 5,924 |
| S4 | 1,001 | 11,870 | 500 | 5,904 | 5,940 |
| S5 | – | – | 500 | 5,964 | 5,945 |
| Total | – | 236,987 | – | 31,456 | 30,382 |
| (b) Conversation Chronicles | | | | | |
| | Train | | Valid | | Test |
| Type | Ep. | Ut. | Ep. | Ut. | Ut. |
| S1 | 160k | 1,730,727 | 20k | 216,950 | 216,171 |
| S2 | 160k | 1,755,788 | 20k | 219,477 | 219,528 |
| S3 | 160k | 1,848,901 | 20k | 230,870 | 230,859 |
| S4 | 160k | 1,977,771 | 20k | 247,993 | 247,501 |
| S5 | 160k | 2,081,149 | 20k | 260,292 | 259,682 |
| Total | 800k | 9,395,436 | 100k | 1,179,582 | 1,171,741 |

Experiments are performed on a large dataset, the MSC. The MSC comprises multi-session conversations between two crowd workers, each embodying predefined personas. In the MSC, the number of history sessions varies from 1 to 4, where session number $i$ indicates that there are $i-1$ prior conversations before the current session. The MSC dataset is publicly accessible through the ParlAI framework at https://parl.ai/projects/msc.

Experiments are also conducted on the Conversation Chronicles dataset (*Jang, Boo & Kim, 2023*) to demonstrate the applicability of our proposed methods to other long-term dialogues. The dataset comprises 1 million multi-session dialogues designed to facilitate long-term conversational research. It incorporates diverse temporal intervals and detailed speaker relationships, enhancing the depth of conversational analysis. The Conversation Chronicles dataset is publicly accessible at https://conversation-chronicles.github.io/ and is distributed under the MIT License.

As a result, we conduct experiments on these two datasets. Since session 1 does not have history conversations, we evaluate our model's performance on session 2–5. Statistics of the MSC dataset and the Conversation Chronicles dataset are summarized in Table 1.

## Implementation details

**Persona generation**. To generate persona, the llama-2-13b-chat model is used without any fine-tuning.

**Persona retrieval**. The bi-encoder, consisting of the persona encoder and the utterance encoder, is initialized from a standard BERT pre-trained model with 12 attention heads and 768 hidden sizes. For the PUP datasets, we train both encoders for up to 40 epochs using Adam (*Kingma, 2014*) optimizer with a learning rate of $10^{-5}$, a batch size of 16, and

a dropout ratio of 0.1. The fine-tuned model is trained with a maximum of three 24GB GPUs (NVIDIA GeForce RTX 3090).

**Response generation**. All baseline models are initialized from a publicly available model. For the MSC and the Conversation Chronicles datasets, we train all baseline models for up to 40 epochs using AdamW (*Loshchilov & Hutter, 2018*) optimizer with a learning rate of $10^{-5}$ and, a batch size of 16. The weight decay for the AdamW optimizer is set to 0.01. GPUs with identical specifications as described above are used.

## Baseline methods

To demonstrate the hypothesis that "summarizing previous dialogues into each speaker's persona while retaining recent dialogues aids response generation", we use the method of concatenating as many dialogues as possible within the model's maximum input length as the baseline and compare it with our proposed GRGPerDialogue Framework. The baseline generation model is described as follows.

- **GPT2**: GPT2 is a transformer-based language model developed by OpenAI, capable of generating human-like text. There are four sizes: GPT2-SMALL, GPT2-MEDIUM, GPT2-LARGE, and GPT2-XL. Their parameters are 124M, 355M, 774M, and 1.5B, respectively. We choose GPT2-MEDIUM, and GPT2-LARGE as baselines.
- **BART**: BART is a transformer-based encoder–decoder model introduced by Facebook AI. It is designed for various natural language processing tasks such as text generation and summarization. There are two different sizes: BART-base and BART-large. Their parameters are 140M and 400M, respectively. We choose BART-base in this study.

## Automatic evaluations

Automatic evaluations used in our analysis are BLEU-n (*Papineni et al., 2002*) and ROUGE-n (*Lin, 2004*). BLEU score serves as a metric for assessing the quality of machine-generated text by comparing it against one or more references. It quantifies the precision of overlapping n-grams between generated text and references. The BLEU-1 metric we use evaluates word-level matching, while BLEU-2 evaluates the matching of consecutive two-word sequences. Similarly, ROUGE is one of the metrics used to evaluate the quality of generated text by comparing it with reference text. A notable distinction from BLEU is that ROUGE measures the count of overlapping n-grams between the generated text and the reference. The ROUGE-L metric employed in our analysis is predicated on the length of the Longest Common Subsequence (LCS) between the generated text and the reference.

## Human evaluations

Evaluating the quality of open-domain dialogue is considered challenging. Reference-based automated evaluation metrics, such as BLEU and ROUGE, can assign low scores even to appropriate responses because they simply compare how much they overlap with actual reference answers (*Liu et al., 2016*). Therefore, human evaluation is desirable to thoroughly verify the quality of conversation across various aspects. To conduct human evaluations, we randomly sample 100 dialogues from the MSC test set. We provide all the sessions and responses generated by each model to four proficient English-speaking annotators. We

ensure that these annotators do not know which model generated which response. We then instruct them to assign scores from 0-4 to the responses based on the following criteria:

- **Fluency** for measuring the smoothness and naturalness of generated responses. Annotators primarily evaluate how smoothly and naturally the model continues the dialogue, checking for grammatical errors or awkward expressions.
- **History relevance** for measuring how well the responses align with the dialogue history. Annotators evaluate whether the model appropriately reflects the context of previous dialogues.
- **Consistency** for measuring the extent to which a model's responses maintain consistency in topics, characters, settings, etc. Annotators evaluate whether the model generates consistent responses in the same topic or situation and whether there are any contradictions or inconsistencies between responses.

## RESULTS

In this section, we present results of both automated evaluation and human evaluation.

### Automatic evaluation

First, we compare baseline models trained by concatenating the dialogue as much as possible with those trained by summarizing previous sessions' dialogue into persona sentences and concatenating only the current session's dialogue. We also compare the use of a pre-defined persona and a generated persona for this summarization. In this experiment, we use the same persona retrieval model trained on the PUP dataset to compare only the effect of persona generation.

**As the dialogue gets longer, summarizing previous sessions' dialogue into each speaker's persona helps with generating responses.** As shown in Table 2, the average token count of dialogues (*e.g.*, Dialogue's Ave. Tokens: 444.84 in Session 2 and 1,598.78 in Session 5) is significantly higher than the token count of pre-defined persona sentences (*e.g.*, Pre-defined Persona Ave. Tokens: 141.39 in Session 2 and 479.49 in Session 5). This indicates that pre-defined persona sentences effectively reduce the dialogue length to approximately one-third, preserving essential information. As demonstrated in Table 3, despite reducing the dialogue length, pre-defined persona sentences still yield comparable or even better performance in longer sessions. For instance, in Session 4, using pre-defined persona sentences with GPT2-medium achieves a BLEU-1 score of 16.13, outperforming the baseline score of 15.35. These results highlight that pre-defined persona sentences are particularly beneficial in mitigating the truncation-induced information loss that occurs in longer dialogues. In our setting (with a model's maximum sequence length of 1,024), most dialogues begin truncating earlier utterances around Session 4. This truncation leads to inevitable information loss in baseline models, whereas pre-defined persona sentences help retain the core context and improve response generation. Consequently, pre-defined persona sentences demonstrate their effectiveness not only in reducing token counts but also in enhancing model performance as dialogues become longer.

**The generated persona is much more powerful than the pre-defined persona.** We find that using the Llama 2 inference ability, we can generate intrinsic persona not explicitly

**Table 2** Average token counts of dialogue, pre-defined persona, and generated persona across different sessions in the MSC dataset.

| Session ID | Dialogue's Ave. Tokens | Pre-defined persona Ave. Tokens | Generated persona Ave. Tokens |
|---|---|---|---|
| Session2 | 444.84 | 141.39 | 156.07 |
| Session3 | 810.19 | 249.30 | 275.47 |
| Session4 | 1,195.08 | 351.65 | 389.88 |
| Session5 | 1,598.78 | 479.49 | 529.35 |

Notes.
The Ave. Tokens for all cases are obtained from either previous session dialogues or generated personas from previous sessions.

**Table 3** Automatic evaluation results on MSC dataset comparing baseline, pre-defined persona, and generated persona settings.

| Model (MSC) | Session 2 | | | Session 3 | | | Session 4 | | | Session 5 | | |
|---|---|---|---|---|---|---|---|---|---|---|---|---|
| | B-1 | B-2 | R-L | B-1 | B-2 | R-L | B-1 | B-2 | R-L | B-1 | B-2 | R-L |
| *b: concatenating the dialogue as much as possible (**b**aseline)* | | | | | | | | | | | | |
| GPT2$_b$-medium | 15.48 | 1.20 | 14.39 | 15.38 | 1.14 | 14.21 | 15.35 | 1.17 | 14.16 | 14.88 | 1.06 | 13.96 |
| GPT2$_b$-large | 16.25 | 1.59 | 15.12 | 16.16 | 1.30 | 15.20 | 15.90 | 1.28 | 14.57 | 15.61 | 1.20 | 14.26 |
| BART$_b$-base | 16.62 | 1.76 | 15.55 | 16.48 | 1.63 | 15.33 | 16.22 | 1.48 | 15.09 | 16.16 | 1.33 | 15.09 |
| *p: replace history with previous sessions' **p**re-defined persona* | | | | | | | | | | | | |
| GPT2$_p$-medium | 15.58 | 1.26 | 14.52 | 15.40 | 1.14 | 14.49 | 16.13 | 1.40 | 15.31 | 15.90 | 1.34 | 14.99 |
| GPT2$_p$-large | 16.26 | 1.50 | 15.28 | 16.14 | 1.52 | 15.26 | 16.47 | 1.62 | 15.65 | 16.30 | 1.54 | 15.48 |
| BART$_p$-base | 16.65 | 1.74 | 15.72 | 16.45 | 1.61 | 15.38 | 16.62 | 1.76 | 15.80 | 16.54 | 1.70 | 15.59 |
| ***Ours**: replace history with previous sessions' generated persona* | | | | | | | | | | | | |
| GPT2$_{Ours}$-medium | 16.55 | 1.61 | 15.19 | 16.14 | 1.50 | 15.38 | 16.56 | 1.40 | 15.24 | 16.42 | 1.56 | 15.13 |
| GPT2$_{Ours}$-large | 16.80 | 1.79 | 15.84 | **17.18** | 1.82 | **16.13** | 17.04 | 1.78 | 16.23 | 16.44 | 1.59 | 15.17 |
| BART$_{Ours}$-base | **17.17** | **1.85** | **15.93** | 17.14 | **1.83** | 16.01 | **17.26** | **1.92** | **16.26** | **17.08** | **1.82** | **15.97** |

Notes.
B-1, B-2, R-L denote BLEU-1, BLEU-2, Rouge-L respectively. Boldface indicates best result for the respective metrics.

present in the dialogue, which significantly aids in response generation. As shown in Table 2, the average token count for generated persona sentences (*e.g.*, 156.07 in Session 2 and 529.35 in Session 5) is consistently higher than that of pre-defined persona sentences (*e.g.*, 141.39 in Session 2 and 479.49 in Session 5). Despite this increase in token count, as can be seen in Table 4, using generated persona resulted in better performance across all models. For instance, when applying generated persona to the BART model, an average improvement of 1% in BLEU-1 score is observed across all sessions compared to baseline models. In our opinion, this is because the generated persona contains richer historical information, such as intrinsic persona. Table 5 shows a case study of the pre-defined persona and generated persona used in the training. The generated persona sentence "*Speaker 1 is currently experiencing hot weather but likes winter*" contains more information than the pre-defined persona sentence "*I like winter*". Additionally, the generated persona sentence above is an intrinsic persona that cannot be obtained solely from Speaker 1's utterance "*Yeah, I like winter, though I do not go outside*", but rather requires inference from the preceding utterance by speaker 2, "*hot enough for you? can not wait for winter*".

**Table 4  Automatic evaluation results on conversation chronicles dataset comparing baseline, pre-defined persona, and generated persona settings.**

| Model (Conversation chronicles) | Session 2 | | | Session 3 | | | Session 4 | | | Session 5 | | |
|---|---|---|---|---|---|---|---|---|---|---|---|---|
| | B-1 | B-2 | R-L | B-1 | B-2 | R-L | B-1 | B-2 | R-L | B-1 | B-2 | R-L |
| *b: concatenating the dialogue as much as possible (**b**aseline)* | | | | | | | | | | | | |
| GPT2$_b$-medium | 16.25 | 1.53 | 15.22 | 15.96 | 1.31 | 15.02 | 15.90 | 1.33 | 14.86 | 15.96 | 1.29 | 14.90 |
| GPT2$_b$-large | 16.57 | 1.61 | 15.60 | 16.46 | 1.54 | 15.20 | 16.29 | 1.44 | 14.98 | 16.14 | 1.42 | 14.28 |
| BART$_b$-base | 16.79 | 1.67 | 15.66 | 16.63 | 1.50 | 15.70 | 16.55 | 1.63 | 15.41 | 16.64 | 1.63 | 15.98 |
| ***Ours**: replace history with previous sessions' generated persona* | | | | | | | | | | | | |
| GPT2$_{Ours}$-medium | 16.40 | 1.49 | 15.26 | 16.05 | 1.32 | 15.10 | 16.22 | 1.44 | 15.25 | 16.08 | 1.40 | 15.11 |
| GPT2$_{Ours}$-large | 16.58 | 1.73 | 15.58 | 16.66 | **1.72** | 15.43 | 16.72 | 1.70 | 15.65 | 16.59 | 1.64 | **16.06** |
| BART$_{Ours}$-base | **16.85** | **1.78** | **15.80** | **16.82** | 1.70 | **15.71** | **16.78** | **1.71** | **15.69** | **16.80** | **1.75** | 15.77 |

**Notes.**
B-1, B-2, R-L denote BLEU-1, BLEU-2, Rouge-L respectively. Boldface indicates best result for the respective metrics.

**Table 5  A case study comparing pre-defined persona and LLM-generated persona based on a sample dialogue from Session 1 of the MSC dataset.**

| The conversation of Session 1 | |
|---|---|
| ... | |
| Speaker 2: | hot enough for you? can not wait for winter. |
| Speaker 1: | Yeah, I like winter, though I do not go outside. |
| ... | |
| Speaker 2: | ... I'm in school. Computer engineering. |
| ... | |
| Speaker 2: | ... I've a real passion for computer programming. |
| Speaker 1: | ... Friends are the best! My vestie bought me a car. |
| **Pre-defined Persona list** | |
| Speaker 1 | I like winter. |
| | My friend bought me a car. |
| Speaker 2 | I like winter. |
| | I am in school for Computer Engineering. |
| **Generated Persona list** | |
| Speaker 1 | is currently experiencing hot weather but likes winter. |
| | has supportive vestie who bought them a car. |
| Speaker 2 | likes winter. |
| | is in school for Computer Engineering. |

**The GRGperDialogue framework is effective even in a conversation where pre-defined personas are not provided.** We also conduct experiments with the Conversation Chronicles dataset, which does not provide a persona. While the experimental setup mirrors that of the MSC dataset, experiments incorporating pre-defined persona are omitted. As shown in Table 4, our proposed framework achieves better results than all baseline models. However, we also observe that the extent of performance improvement is generally lower than when applying the MSC dataset. We believe that this is because, while constructing the

**Table 6  Automatic evaluation results on MSC dataset comparing models with and without persona retrieval.**

| Model (MSC) | Session 2 | | | Session 3 | | | Session 4 | | | Session 5 | | |
|---|---|---|---|---|---|---|---|---|---|---|---|---|
| | B-1 | B-2 | R-L | B-1 | B-2 | R-L | B-1 | B-2 | R-L | B-1 | B-2 | R-L |
| *wo: WithOut retrieval process* | | | | | | | | | | | | |
| GPT2$_{wo}$-medium | 16.23 | 1.59 | 15.39 | 16.08 | 1.36 | 15.04 | 15.44 | 1.20 | 14.26 | 14.90 | 1.06 | 13.82 |
| GPT2$_{wo}$-large | 16.82 | 1.74 | 15.76 | 16.74 | 1.68 | 15.66 | 16.57 | 1.70 | 15.61 | 16.09 | 1.36 | 15.14 |
| BART$_{wo}$-base | 16.98 | 1.81 | 15.88 | 16.84 | 1.76 | 15.85 | 16.68 | 1.70 | 15.78 | 16.32 | 1.55 | 15.39 |
| *Ours: concatenating retrieved persona sentences from the generated persona list with current session's utterance* | | | | | | | | | | | | |
| GPT2$_{Ours}$-medium | 16.55 | 1.61 | 15.19 | 16.14 | 1.50 | 15.38 | 16.56 | 1.40 | 15.24 | 16.42 | 1.56 | 15.13 |
| GPT2$_{Ours}$-large | 16.80 | 1.79 | 15.84 | **17.18** | **1.82** | **16.13** | 17.04 | 1.78 | 16.23 | 16.44 | 1.59 | 15.17 |
| BART$_{Ours}$-base | **17.17** | **1.85** | **15.93** | 17.14 | **1.83** | 16.01 | **17.26** | **1.92** | **16.26** | **17.08** | **1.82** | **15.97** |

**Notes.**
B-1, B-2, R-L denote BLEU-1, BLEU-2, Rouge-L respectively. Boldface indicates best result for the respective metrics.

MSC dataset, two crowd workers generated conversations based on a pre-defined persona, resulting in conversations containing many persona elements.

Additionally, experiments are conducted to assess effect of the persona retrieval on response generation. In these experiments, we compare responses generated by concatenating all personas with those generated only with the retrieved persona.

**The process of retrieving a persona relevant to the utterance helps generate responses.** As seen in Table 6, the retrieval process shows better results in all cases. This means that it is more advantageous to utilize the persona retrieval model rather than relying solely on the inference ability of the generation model when generating responses. In particular, we find that performance decreases without using the retrieval process as the session increases. We believe this is because, as the number of sessions increases (an average of five persona sentences of each speaker per session), the amount of generated persona information also increases, potentially leading the model to become confused about which persona sentences to use when generating responses.

**The retrieval model trained on the PUP dataset effectively retrieves persona sentences relevant to the current utterance.** A study on the Modular Prompted Chatbot (*Lee et al., 2023*) utilized a DPR model, named MultiDoc2Dial (*Feng et al., 2021*), to retrieve historical dialogue information relevant to the current utterance. However, the MultiDoc2Dial model was trained on pairs of Wikipedia passages and utterances, making it unsuitable for retrieving relevant persona sentences. To evaluate our proposed retrieval model, we compared the results of the MultiDoc2Dial model with our retrieval model. Initially, we examine persona sentences retrieved by each retrieval and their similarity scores. Our findings indicate that both models exhibit a degree of proficiency in persona retrieval. However, as seen in Table 7, the retrieval model proposed in a previous study (*Feng et al., 2021*) assigns high scores even to unrelated persona, posing a challenge in setting a threshold because of the narrow score margin between related persona and unrelated persona. Unlike MultiDoc2Dial, where the similarity scores are narrowly distributed and make thresholding difficult, our model produces more distinguishable similarity scores between relevant and irrelevant persona candidates. Based on qualitative analysis, we

**Table 7  A case study on the persona retrieval model we proposed, compared to the MultiDoc2Dial model.**

| Utterance | Speaker: I started a new diet that should help with my blood sugar. |
|---|---|
| Persona List | |
| 1. Speaker is read for an exam. | |
| 2. Speaker has a busy schedule with college and no time for school. | |
| 3. Speaker enjoys eating desserts and gained weight during the holidays. | |
| ... | |
| 9. Speaker is looking forward to outdoor activities in the spring to lose weight. | |
| Top 3 retrieved persona sentences and similarity scores | |
| MultiDoc2Dial | (speaker enjoys eating desserts and gained weight during the holidays, 69) |
| | (speaker is looking forward to outdoor activities in the spring to lose weight, 68) |
| | (speaker is reading for an exam, 66) |
| Persona retrieval (ours) | (speaker enjoys eating desserts and gained weight during the holidays, 72) |
| | (speaker is looking forward to outdoor activities in the spring to lose weight, 71) |
| | (speaker is reading for an exam, 60) |

**Table 8  BLEU-1 scores for different persona similarity thresholds on Session 4.**

| Threshold | 0.60 | 0.62 | 0.64 | 0.66 | 0.68 | 0.70 |
|---|---|---|---|---|---|---|
| BLEU-1 (Session 4) | 16.97 | 17.03 | 17.05 | 17.12 | 17.26 | 17.22 |

observed that scores below 0.6 often corresponded to weakly related or irrelevant persona sentences, while scores above 0.7 sometimes excluded valid persona sentences. We therefore focused on the 0.6–0.7 range and evaluated thresholds in increments of 0.02. Based on BLEU scores, we selected 0.68 as the optimal threshold. To illustrate the impact of the threshold on generation performance, we report BLEU-1 scores from Session 4, where the score differences across thresholds were most prominent. As shown in Table 8, the performance peaks at 0.68, supporting our choice of threshold.

## Human evaluation

In addition to automatic evaluation, we conduct human evaluation on the responses generated by BART-base. The reason for this is that BART achieves the highest performance in automated evaluation of generated responses. Results of the human evaluation are summarized in Table 9. Generally, our proposed method outperforms all baseline methods across all perspectives. In terms of fluency, all models demonstrate similar performance, which means BART models can generate fluent and grammatically correct responses. In terms of history relevance, our method achieves the best performance. The overall scores, however, are relatively low, primarily because only a small portion of responses in the dataset require historical context. Many responses—such as greetings or topic-initiating utterances—do not rely on prior context, leading annotators to assign lower scores in

**Table 9  Human evaluation of the response generation by different methods.**

| Model | Fluency | History relevance | Consistency |
| --- | --- | --- | --- |
| $BART_b$ | 2.23 | 1.11 | 2.68 |
| $BART_p$ | 2.18 | 1.21 | 2.73 |
| $BART_{Ours}$ | **2.29** | **1.48** | **2.82** |

**Notes.**
  The highest BLEU-1 score is highlighted in bold.

those cases. Similarly, our method achieves the best result in consistency, indicating that generated responses are non-contradictory.

### Qualitative example

Table 10 presents a qualitative case study demonstrating how generated and retrieved personas influence BART's response generation. The example involves a dialogue where the speaker mentions majoring in computer engineering, having a passion for programming, and aspiring to start a company with a friend.

This case is analyzed from the perspectives of fluency, history relevance, and consistency. In terms of fluency, all model responses are grammatically correct and natural, showing no significant difference across conditions. However, notable differences emerge in history relevance. Without persona information, the model generates a generic response such as "just a normal day at school," which lacks contextual relevance to the current question, "How is school going?" When using only the generated persona, the model includes unrelated information such as "starting a company," which does not directly answer the question. In contrast, combining generated and retrieved personas results in a response that accurately reflects the context, referencing "time on the computer" and "engineering assignments," which are more aligned with the speaker's current situation. A similar trend is observed in consistency. The use of only generated personas may introduce off-topic content that weakens coherence, while incorporating retrieved personas helps the model stay more aligned with the speaker's established identity and earlier utterances. This example illustrates the limitations of relying solely on generated persona—which can introduce irrelevant details—and the effectiveness of combining it with retrieved persona to improve both the history relevance and consistency of generated responses.

## CONCLUSION

In this work, we present a framework with GRGPerDialogue that can generate persona from history conversations, retrieve persona relevant to the current utterance, and generate responses. The persona generator dynamically generates each speaker's persona from previous sessions of conversations using Llama 2 in-context learning. After that, Persona Retrieval, trained with the PUP dataset, retrieves persona sentences relevant to the current utterance. Finally, dialogue generation models such as GPT2 and BART generate responses using retrieved persona sentences and current session conversations. Experimental results indicate that replacing old history with persona outperforms all baseline models across various metrics. This, suggests that our method is effective for response generation in

**Table 10  A qualitative case study illustrating how the use of generated and retrieved personas influences BART-generated responses.**

| Previous dialogue | |
|---|---|
| *(Excerpt from Session 1; later dialogue omitted for brevity)* | |
| Speaker 1: | Good morning, then! Hot enough for you? Can't wait for winter. |
| Speaker 2: | Yeah, I like winter, though I don't go outside much anyway. |
| Speaker 1: | Same here, it's a little hard with my wheelchair. |
| Speaker 2: | Oh, I imagine so! I'm in school. Computer engineering. |
| Speaker 1: | Interesting! With cooler weather coming you gotta watch out for colds. |
| Speaker 2: | Yeah, that's true. I've a real passion for computer programming. |
| Speaker 1: | Friends are the best! My vestie actually bought me a car last year. |
| | ... |
| **Generated Persona (Speaker 2)** | |
| - likes winter. | |
| - is in school for computer engineering. | |
| - has a passion for computer programming. | |
| - hopes to start a company with their best friend. | |
| **Retrieved Persona (Speaker 2)** | |
| - spends a lot of time on the computer. | |
| - majors in engineering. | |
| **Current Utterance** | Speaker 1: How is school going? |
| **Ground Truth (GT)** | Speaker 2: Going well, on my computer all the time. How are you? |
| **Model w/o Persona** | Speaker 2: It's going okay. Just a normal day at school. |
| **Model w/Generated Persona** | Speaker 2: School is good! I've been thinking a lot about starting a company with my best friend. |
| **Model w/Gen. + Retrieved Persona** | Speaker 2: It's going great! I've been on my computer all the time working on engineering assignments. |

long-term dialogue. According to human evaluation results, our method's responses are perceived to be more natural, history-relevant, and consistent than baseline models' responses.

## LIMITATIONS AND FUTURE WORKS

In the GRGPerDialogue framework, we utilize the in-context learning capability of Llama 2 to generate persona from previous sessions and persona from utterances to generate the PUP dataset. However, persona sentences generated by Llama 2 still suffer from hallucination and factual inaccuracy. To better understand these limitations, we manually analyzed 100 randomly sampled persona sentences and categorized the error types. Specifically, we observed that 1% of the samples included persona sentences incorrectly attributed to external individuals mentioned in the conversation, while 3% contained sentences that

could not be inferred from the dialogue at all. As part of future work, we plan to further analyze such cases, correct the erroneous samples, and fine-tune Llama 2 with the refined data to improve persona generation accuracy.

Another limitation is that the quality of the generated responses ultimately depends on the underlying generation model. Specifically, it relies on well-trained, large pre-trained generation models to achieve optimal performance. However, due to the limitations of available GPU resources, this study did not utilize larger models such as Llama 2 for training. Instead, we conducted experiments using GPT2 and BART models. While our approach demonstrated performance improvements over baseline methods, it did not outperform state-of-the-art large language models (LLMs). This highlights the dependency on model size and capacity for further improvements, suggesting that future work should involve leveraging larger generation models to fully explore the potential of our framework.

### Funding

This research was supported by the MSIT (Ministry of Science and ICT), Korea, under the ITRC (Information Technology Research Center) support program (IITP-2025-2020-0-01789), and the Artificial Intelligence Convergence Innovation Human Resources Development (IITP-2025-RS-2023-00254592) supervised by the IITP (Institute for Information & Communications Technology Planning & Evaluation). The funders had no role in study design, data collection and analysis, decision to publish, or preparation of the manuscript.

### Grant Disclosures

The following grant information was disclosed by the authors:
The MSIT (Ministry of Science and ICT), Korea, under the ITRC (Information Technology Research Center) support program: IITP-2025-2020-0-01789.
The Artificial Intelligence Convergence Innovation Human Resources Development, supervised by the IITP (Institute for Information & Communications Technology Planning & Evaluation): IITP-2025-RS-2023-00254592.

### Competing Interests

The authors declare there are no competing interests.

### Author Contributions

- Dohyun Cha conceived and designed the experiments, performed the experiments, analyzed the data, performed the computation work, prepared figures and/or tables, authored or reviewed drafts of the article, and approved the final draft.
- Dawon Lee performed the experiments, analyzed the data, performed the computation work, authored or reviewed drafts of the article, and approved the final draft.
- Jihie Kim conceived and designed the experiments, authored or reviewed drafts of the article, and approved the final draft.

## Data Availability

Code and raw data are avaiable at GitHub and Zenodo:

- https://github.com/cha970214/GRGPerDialogue

- cha970214. (2025). cha970214/GRGPerDialogue: GRGPerDialogue-PUP dataset (v1.0). Zenodo. https://doi.org/10.5281/zenodo.15363966

The Multi-Session Chat Dataset is available at ParlAI: https://parl.ai/projects/msc

The Conversation Chronicles dataset is available at GitHub: https://conversation-chronicles.github.io/.

## Supplemental Information

Supplemental information for this article can be found online at http://dx.doi.org/10.7717/peerj-cs.2979#supplemental-information.

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
