# Peer review of "Generating, retrieving persona and generating responses for long-term open-domain dialogue"

_PeerJ Computer Science, doi:10.7717/peerj-cs.2979_

## Round 0.1 · original submission · Minor Revisions

Dear Authors,

Your manuscript is close to acceptance, but the reviewers request minor revisions:

1. Method clarity – Distinguish clearly between dialogue-history summaries and persona summaries; justify the use of manual summaries and the 0.68 similarity threshold; specify how irrelevant sentences are filtered.

2. Qualitative insight & hallucination check – Add a brief error analysis plus a quantitative measure of persona factuality, and include a few illustrative examples contrasting your model with the baselines where long-term context matters.

3. Scope & comparison – Comment on generalisation to other domains/datasets, explain the impact when few responses need history, and position your method against current state-of-the-art LLMs.

4. Presentation – Expand table/figure captions and fix minor typographical issues.


Best regards,

Reviewer 1 ·

Basic reporting

1. Strengths
• Relevance and Innovation: The study addresses the challenge of long-term open-domain dialogue generation, a timely and complex problem, proposing a novel GRGPerDialogue framework.
• Framework Design: The three-step structure (persona generation, retrieval, and response generation) offers a well-motivated and comprehensive solution to context loss in long-term conversation.
• Evaluation: The use of both automatic metrics (BLEU, ROUGE) and human evaluation adds robustness to the results.
• Dataset Contribution: Introduction of the Persona-Utterance Pair (PUP) dataset is a useful addition for future research on persona retrieval.

2. Weaknesses
• Limited Generalization: While the framework is tested on two datasets, the performance on Conversation Chronicles is relatively modest, and further testing across diverse languages and domains is encouraged.
• Dependence on Llama 2: The framework's reliance on Llama 2 for persona generation raises concerns about hallucinations and factual accuracy, which are acknowledged but not quantitatively addressed.
• Computational Cost: The system relies on multiple large-scale models, which may limit real-world deployment in resource-constrained settings.


3. General Comments for the Authors
• The manuscript is written in clear and professional English.
• Literature is well-cited and provides appropriate context.
• The overall manuscript is well-written, with clearly presented methodology and results. However, some figures (e.g., Figure 3) could benefit from more detailed annotations and labeling to guide interpretation.
• Consider explaining how the threshold for persona similarity (0.68) was selected in greater detail, possibly with visualization of performance across thresholds.
• The ablation study is informative, but adding more qualitative examples comparing outputs from baseline and proposed models would strengthen the empirical claims.
• The paper may benefit from a brief error analysis section, identifying typical failure cases or hallucination patterns from generated personas.

Experimental design

The research is original and fits the scope of PeerJ Computer Science.
The hypothesis is clearly defined and well-justified.
Experiments are replicable, with sufficient detail on datasets, models, and training procedures.
The introduction of the PUP dataset and its application in DPR model training is a notable contribution.

Validity of the findings

Results are statistically and methodologically sound.
The use of baseline comparisons, multiple sessions, and both pre-defined and generated personas strengthens the findings.
However, performance gains are modest (~0.6–1%), which, while consistent, may benefit from deeper qualitative insights.
Conclusions are well-supported and appropriately scoped.

Additional comments

Recommendations
1. Add a Quantitative Hallucination Evaluation: Given the known risk of hallucinations in LLMs, an evaluation measuring the factual correctness of generated personas would improve reliability.
2. Expand Evaluation to Diverse Domains: Applying the framework to other long-term dialogue datasets in different domains (e.g., customer support, healthcare) would demonstrate broader applicability.

·

Basic reporting

- page 2: line 48 you mention summarization methods to condense the dialogue history. In line 62 you mention that you also summarize previous history as persona. Please make more clear what the difference is. Is this just that the history spans multiple dialogues or is there also a difference in content.

- page 3, line 114why are manual summaries required? They can also be created automatically. Please make clear why manual. In line 117 you refer to Want et al 2023 that summarise automatically. Why is it a problem that they need to be verified for hallucination? In your limitation section you also admit that your approach suffers from hallucinations.
P.6, line 215/116: irrelevant sentences or words are removed. How is this decided? Manually? There are way too many sentences to do this manually.

Captions of tables could be more elaborate.

Experimental design

The experimental design is solid and good work.

-

Validity of the findings

P12, Line 401: only a low proportion of responses require historical information. How is this decided? Did the authors take a sample? If this is true, how relevant is this work? Should you test on another data set in which this is more relevant?

In most cases only a little improvement is made in relation to the baseline (1%). This also maske me wonder how relevant this work is or whether a different data set should be used. It is a pity that the results section focuses on performance and a deeper error analysis is lacking. I suggest that the authors select cases where the history does matter and try to assess the errors and good responses of the approach. This would give more clear insights how to improve, especially as the human evaluation scores are rather low on a range from 1 to 5.

Finally, the limitations section mentions that state-of-the-art LLM perform a lot better. I would like to know what their performance is and whether the proposed method can handle cases that the SOTA LLMs cannot (e.g. in case of very rich histories).

Additional comments

Overall a good paper and interesting approach with potential. It is a pity that only few test cases require historical context. Diving into these cases may make the paper more convincing.

---

## Round 0.2 · accepted · Accept

The authors have addressed all of the reviewers' comments. Congrats.

Reviewer 1 ·

Basic reporting

The revised manuscript demonstrates a clear improvement in structure, clarity, and language quality.
All previously noted issues have been addressed effectively

Experimental design

All concerns regarding experimental design have been resolved.

Validity of the findings

The validity of the findings is now well-supported and complete.

Additional comments

No further revisions are needed at this stage.